# Papain Affects the Percentage and Morphology of Microglia in Hippocampal Neuron–Glial Cultures

**DOI:** 10.3390/brainsci15050442

**Published:** 2025-04-24

**Authors:** Ivan A. Tumozov, Valentina N. Mal’tseva, Sergei A. Maiorov, Artem M. Kosenkov, Sergei G. Gaidin

**Affiliations:** Federal Research Center “Pushchino Scientific Center for Biological Research of the Russian Academy of Sciences”, Institute of Cell Biophysics of the Russian Academy of Sciences, 142290 Pushchino, Russia; ivantumoz@mail.ru (I.A.T.); mvn3@mail.ru (V.N.M.); dikyagux@mail.ru (S.A.M.); kosenkov_am@pbcras.ru (A.M.K.)

**Keywords:** microglia, triculture, papain, CDr20, intracellular Ca^2+^ concentration, Iba1, ATP

## Abstract

**Background.** Microglia, accounting for 5–15% of total brain cells, represent an essential population of glial cells in the cultures used for modeling neuroinflammation in vitro. However, microglia proliferation is poor in neuron–glial cultures. Here, we studied the population composition of rat hippocampal neuron–glial cell cultures prepared utilizing papain (PAP cultures) and trypsin (TRY cultures) as proteolytic enzymes for cell isolation. **Methods.** To evaluate the percentage and morphology of microglia in TRY and PAP cultures and cultures incubated in the presence of TGFβ+MCSF+cholesterol, which should enhance microglia proliferation, we used an immunostaining and calcium imaging approach in combination with staining using the recently developed vital microglia fluorescent probe CDr20. **Results.** We have shown that the microglia percentage in PAP cultures was higher than in TRY cultures. Microglia in PAP cultures are predominantly polarized, while bushy morphology was more characteristic of TRY cultures. We have also demonstrated that the TGFβ+MCSF+cholesterol combination increases the microglia number both in PAP and TRY cultures (up to 25–30%) and promotes the appearance of ameboid microglia characterized by high mobility. However, the significant appearance of ameboid microglia was observed already at the early stages of cultivation (2 DIV) in TRY cultures, while in PAP cultures, the described transformation was observed at 7 DIV. Based on the absence of the ATP-induced Ca^2+^ response, round shape, significant proliferation, and high mobility, we have suggested that ameboid microglia are reactive. **Conclusions.** Thus, our results demonstrate that papain is a more suitable proteolytic enzyme for preparing mixed hippocampal neuron–glial cultures with a higher percentage of heterogeneous microglia and functional neurons and astrocytes (tricultures).

## 1. Introduction

According to modern neurobiology paradigms, microglia are one of the main populations of glial cells in the mammalian brain. These cells function as resident brain macrophages and play a crucial role in neuroinflammatory processes [1]. Through their involvement in synaptic pruning (the elimination of excess synapses), microglia also serve as key regulators of synaptic plasticity [2]. The microglial population exhibits inherent heterogeneity, further amplified under pathological conditions due to phenotypic transformations [1]. Primary microglia cell cultures [3] or immortalized microglia-like cell lines [4] are a convenient and common object for in vitro neuroinflammation research. However, modeling comprehensive inflammatory responses necessitates intercellular interactions among diverse cell types [3]. To overcome this limitation, approaches are being developed to obtain co-cultures of separately grown populations of brain cells [5]. Although this approach makes it possible to vary the experimental design flexibly—namely, by choosing the moment of addition of one cell population to another, the number of cells introduced, and pretreatment of one of the groups of cells with various test compounds—it is not without disadvantages. In addition to methodological difficulties in obtaining pure cell cultures of a particular population of brain cells and their different maturation periods—often requiring mixing of cells of different passages (or even from different animals)—the absence of microenvironment in pure neuronal, microglial, or astrocytic cultures, as well as the manipulations aimed at “purification”, such as the addition of cytosine arabinoside in the case of neuronal cultures [6,7,8], or prolonged intense shaking in the case of astrocytic cultures [9], affect the phenotype of cells in the obtained cultures.

Mixed tricultures in which the formation of closed, self-sufficient neuron–glial networks occurs and cells, including microglia, proliferate and mature in a more in vivo approximate microenvironment, can be proposed as a solution to this problem. However, established protocols for mixed neuron–glial cultures utilizing trypsin for extracellular matrix degradation and Neurobasal medium+B27 Supplement combination as a culture medium typically result in limited microglial proliferation [10]. The addition of growth factors has been proposed to increase the number of microglia in mixed neuron–glial cultures, but a similar approach has been reported only in several studies [10,11,12].

Papain is often used as a digestive agent during the preparation of primary microglia cultures [5,13]. Although both papain and trypsin are proteolytic enzymes, the former belongs to cysteine proteases, while the latter is a serine protease. This difference means that the hydrolysis of polypeptide sequences mediated by trypsin and papain occurs at different sites. Despite the long history of using these proteases during brain cell culture preparation, their effects were compared only in several studies [14,15] and only in the context of neurons. Based on these facts, we have decided to test how papain affects the functional state and population composition of the cultures, particularly the number of microglia.

In this investigation, we examined how papain utilization for extracellular matrix degradation affects the population composition of mixed rat hippocampal neuron–glial cultures (PAP cultures). Our findings demonstrate that papain use promotes increased microglial populations compared to trypsin-prepared cultures (TRY cultures) without significant alterations in astrocyte and neuron quantities. Additionally, we observed that microglia in PAP cultures predominantly exhibit polarized morphology, whereas TRY cultures display predominantly bushy microglial morphology.

## 2. Materials and Methods

### 2.1. Preparation of Cell Cultures

The hippocampi of newborn male Wistar rats (P0-2) were used to prepare cell cultures as described previously [16]. The brain extracted after decapitation was placed in a Versene ice-cold solution, and the hippocampus was separated. Then, the hippocampal tissue was cut into small fragments with scissors. Afterward, the Versene solution was replaced with a 1% trypsin solution (in Versene solution) or a 0.8% papain solution (in Versene solution). As in our numerous previous studies [17,18,19,20,21,22], incubation with trypsin was carried out at 37 °C for 10 min under constant shaking (500 rpm) in a thermoshaker. In the case of papain, the incubation time was 25 min, as ineffective matrix dissociation occurred with less incubation time, and the cell suspension contained large amounts of debris. Hippocampal fragments treated with these enzymes were washed twice with cold Neurobasal medium and then gently triturated in 1 mL of Neurobasal medium with an automatic pipette (a 1 mL tip was used). Undissociated tissue fragments were carefully removed by pipette, and the obtained suspension was centrifuged for 3 min at 2000 rpm. The resulting cell pellet was resuspended in Neurobasal medium to obtain a suspension with a cell density of 700,000 cells/mL. The Neubauer counting chamber (JVLAB, Foshan, China) was used to evaluate the quantity of cells before the dilution. The number of dead cells stained with 0.4% Trypan Blue solution did not exceed 1–2 cells per 5 squares of the chamber in the case of both papain and trypsin. Glass cylinders with an inner diameter of 6 mm were used to concentrate the cells on the confined surface. The cylinders were set on round cover glasses (VWR International, Radnor, PA, USA) of 24 mm diameter placed in sterile Petri dishes of 35 mm diameter and pre-coated to improve cell adhesion with polyethyleneimine solution (1 mg/mL) for 50 min, as described previously [16], followed by overnight drying. A total of 100 μL of suspension was poured into each cylinder. Petri dishes with cylinders filled with cell suspension were covered with lids and carefully transferred to a CO_2_ incubator for 40 min for cell attachment. After that, the cylinders were carefully removed using tweezers, and 2 mL of the culture medium consisting of Neurobasal medium, 2% serum-free B27 supplement, 0.5 mM glutamine, and penicillin/streptomycin (100 U/mL) was added to the dishes. The density of cells in the cultures obtained in accordance with this procedure is approximately 50,000 cells/mm^2^. Since postnatal cells were used to prepare cultures, sterile sodium chloride solution was added to Neurobasal medium to achieve a final NaCl concentration in the medium of 4 g/L. The cells were grown in a CO_2_ incubator at 37 °C, 95% humidity, and 5% CO_2_ in the atmosphere. Cells were taken for experiments on the 2nd, 7th, and 14th days of culture (2, 7, and 14 DIV, respectively). In the case of cultures grown using additionally introduced factors (TGFβ 2 ng/mL, MCSF 5 ng/mL, cholesterol 100 μg/mL), the addition of compounds to the culture medium was carried out immediately after removal of the cylinders.

### 2.2. Fluorescent Microscopy

#### 2.2.1. Calcium Imaging

Fluorescence microscopy was used to evaluate the population composition of cell cultures and the physiological activity of cells. Identification of neurons and glial cells was carried out based on calcium response (an increase in intracellular Ca^2+^ concentration ([Ca^2+^]_i_)) [17,18] to the addition of potassium chloride (35 mM) and adenosine triphosphate (ATP, 10 μM; this concentration is sufficient to activate most of subtypes of P_2_X and P_2_Y receptors [23]), respectively. For this purpose, the cells were stained with a fluorescent Ca^2+^-sensitive probe FluoriCa-8 AM (Fluo-8 AM analog). The probe staining and all subsequent experiments were performed in Hank’s balanced salt solution (HBSS) of the following composition: 136 mM NaCl, 3 mM KCl, 0.8 mM MgSO_4_, 1.25 mM KH_2_PO_4_, 0.35 mM Na_2_HPO_4_, 1.4 mM CaCl_2_, glucose 10 mM; HEPES 10 mM, pH = 7.35. FluoriCa-8 was loaded at a temperature of 28 °C for 40 min, after which the cultures were washed twice with HBSS and washed again 10 min later to allow the unesterified dye to leave the cells. The final concentration of FluoriCa-8 in the working solution was 5 μM. After washing, the coverslips with cell cultures were mounted in a microscopic chamber and placed on an inverted epifluorescence microscope Leica 6000 DMI (Leica Microsystems, Wetzlar, Germany), equipped with Leica IL-6000 illuminator (mercury lamp) (Leica Microsystems, Wetzlar, Germany) and a Hamamatsu 9100C CCD camera (Hamamatsu Photonics K.K., Hamamatsu City, Japan). An L5 filter cube with BP 480/40 excitation filter, a 505 nm dichroic mirror, and a BP 527/30 emission filter was used for the excitation and emission detection of FluoriCa-8. The short-term addition of KCl and ATP and the subsequent washout were carried out in a flow of HBSS using a special perfusion system (flow rate 10 mL/min). In calcium imaging experiments, where more than one dye was not required, the fluorescent ratiometric Ca^2+^-sensitive probe Fura-2 AM was used to assess changes in [Ca^2+^]_i_. Loading of cells with Fura-2 (3 μM) was carried out under conditions similar to those of FluoriCa-8 loading. Fura-2 cube, the characteristics of which are given above, was used to excite and record the emission.

#### 2.2.2. Vital Microglia Identification

A fluorogenic substrate of UDP-glucoronosyl transferase [24] CDr20 Microglia Stain (CDr20) was used to identify microglia cells. Since the enzyme is specific for microglia cells, other cell types do not transform the dye in culture. A TX2 (Leica Microsystems, Wetzlar, Germany) filter cube with BP 560/40 excitation filter, a dichroic mirror 595 nm, and a BP 645/75 emission filter was used to excite and record the CDr20 fluorescence. CDr20 application was performed in HBSS flow, and the dynamics of the staining were registered with a frame rate of 1 frame per 5 s. A double-stranded DNA-binding fluorescent dye, Hoechst 33342, at a concentration of 5 μg/mL, was used to visualize the nuclei to further count the total number of cells in the culture. The cells were stained with the dye for 10 min and then washed with HBSS. An external filter BP387/15 and Fura-2 filter cube with 72100bs dichroic mirror and HQ 540/50m emission filter were used to excite and collect the Hoechst 33342 emission.

### 2.3. Immunocytochemical Staining

Immunocytochemical staining was used to identify neurons, astrocytes, and microglia in rat hippocampal cell cultures, as well as to assess the presence of glutamatergic and GABAergic synapses on neurons. Cell cultures removed from the CO_2_ incubator were washed three times with phosphate-buffered saline (PBS; pH = 7.35) and then fixed with a 4% solution of freshly prepared paraformaldehyde for 20 min. The paraformaldehyde solution was prepared by dissolving the weighed amount in PBS with constant stirring at 70 °C in a fume hood until a clear solution was obtained. After cooling the solution, undissolved aggregates were removed by filtration through a polyethersulfone membrane with a pore size of 0.22 μm. Cells were washed with ice-cold PBS three times to remove paraformaldehyde. Goat serum (10% in PBS) was used to block the nonspecific binding of secondary antibodies. Simultaneously with blocking, cell membrane permeabilization was performed using Triton X-100 at a concentration of 0.1% for intracellular antigens. In contrast, a milder detergent, Tween 20, was used for surface antigens at a concentration of 0.2%. Goat serum and detergents were dissolved in PBS. Cells were incubated with a blocking solution containing the corresponding detergent at room temperature for 30 min, after which cells were washed once with 1% goat serum solution (in PBS) for 5 min. Primary antibodies were dissolved in the same solution. The dilutions of primary antibodies are indicated in the corresponding figure legends. Cells were incubated with primary antibodies at +4 °C for 12 h, followed by triple washing with PBS (each PBS incubation interval lasted 5 min). Then, cell cultures were incubated with goat secondary antibodies dissolved in PBS for 90 min in the dark at room temperature, followed by triple washing with PBS. Secondary antibodies used were goat anti-mouse immunoglobulins (conjugated with Alexa Fluor 647) and anti-rabbit immunoglobulins (conjugated with Alexa Fluor 488). Secondary antibodies were diluted 1:300. To visualize cell nuclei, cells were incubated with Hoechst 33342 (5 μg/mL in PBS) for 10 min and washed twice with PBS.

The distribution of antibodies to synaptic markers vGAT and vGluT2 was visualized using a Leica TCS SP5 (Leica Microsystems, Wetzlar, Germany) confocal microscope. Neuron-specific enolase (NSE) served as a neuronal marker. Goat antibodies to mouse immunoglobulins (conjugated to Alexa Fluor 647), rabbit immunoglobulins (conjugated to Alexa Fluor 555), and guinea pig immunoglobulins (conjugated to Alexa Fluor 488) were used as secondary antibodies. The dilution of all secondary antibodies was 1:300. Dilutions of primary antibodies are indicated in the figure legends. He-Ne lasers with excitation bands 633 and 543 nm and an argon laser with excitation band 488 nm were used for excitation. Scanning was performed in Z-stack mode, with each dye being scanned in a separate stack with one active laser to avoid overlap of excitation/emission spectra.

### 2.4. Scratch Assay

A scratch wound was made by scraping the cell monolayer across the cover glass with a sharp sterile syringe needle. The cultures were incubated for 48 h after the scratching and then stained with antibodies against GFAP and Iba1. For the experiments, 7 DIV cultures were taken. The width of the scratch averaged 300 μm.

### 2.5. Data Analysis

Image processing was performed using ImageJ version 1.53s (NIH, Bethesda, MD, USA) and Leica Application Suite X version 3.7.6.25997 (Leica Microsystems, Wetzlar, Germany) software. The total number of cells in the field of view was counted in ImageJ using the Analyze Particles software option. Images of Hoechst 33342-stained cell nuclei were used for counting. Due to the peculiarities of cell cultures, consisting of the formation of conglomerates, cells in these structures were counted manually to avoid artefacts caused by the peculiarities of the Analyze Particles algorithm. OriginLab Pro 2021 version 9.8.0.200 (OriginLab, Northampton, MA, USA) software was used for plotting graphs. Statistical analysis of data and diagram construction were performed using GraphPad Prism 8 version 8.0.2 (GraphPad Software, San Diego, CA, USA). Data are presented as mean ± SD or as median ± interquartile range in the case of non-parametric tests. The normality of data was evaluated with the Shapiro–Wilk test. The number of independent cell culture preparations used in experiments is marked as “N”, whereas the total number of analyzed cells (in all used preparations) is marked as “n”. As a rule, cell culture preparations from 4 to 5 different animals were used in experiments.

### 2.6. Substances

The reagents that were used in the experiments are listed below. (1) Sigma-Aldrich, Saint Louis, MO, USA: Poly(ethyleneimine) solution (Cat. no. P3143), TWEEN^®^ 20 (Cat. no. P1379), Paraformaldehyde (P6148); L-Glutamine (Cat. no. G8540), Anti-SLC32A1 (N-terminal) antibody (anti-vGAT) produced in rabbit (Cat. no. SAB2700790). (2) Life Tech-nologies, Grand Island, NY, USA: B-27 supplement (Cat. no. 17504044), Trypsin 2.5% (Cat. no. 15090046). (3) Molecular Probes, Eugene, OR, USA: Fura-2 AM (Cat. no. F1221). (4) Cayman Chemical, Ann Arbor, MI, USA: Bicuculline (Cat. no. 11727). (5) Paneco, Moscow, Russia: Neurobasal medium (H333), penicilin-streptomycin (A063). (6) Abcam, Cambridge, UK: Goat Anti-Mouse IgG H&L (Alexa Fluor^®^ 647) (Cat. no. ab150115), Goat Anti-Rabbit IgG H&L (Alexa Fluor^®^ 555) (Cat. no. ab150078), Goat Anti-Guinea pig IgG H&L (Alexa Fluor^®^ 488) (Cat. no. ab150185), Goat serum New Zealand origin (Cat. no. 16210072). (7) HyTest, Moscow, Russia: mouse monoclonal antibodies to neuron-specific enolase (Cat. no. 4N6), mouse monoclonal antibodies to glial fibrillar acidic protein (Cat. no. 4G25). (8) Lumiprobe, Moscow, Russia: FluoriCa-8 AM (Fluo-8 AM analog) (Cat. no. 3391), LumiCell CDr20 Microglia Stain (Cat. no. 3570). (9) Sci-Store, Moscow, Russia: active human macrophage colony-stimulating factor (MCSF) (Cat. no. PSG280). (10) Cloud-Clone Corp, Wuhan, China: Polyclonal Antibody to Allograft Inflammatory Factor 1 (AIF1) (Cat. no PAC288Ra01), Active Transforming Growth Factor Beta 1 (Cat. no. APA124Ra01). (11) CDH Ltd., New Delhi, India: Cholesterol (Cat. no. 043011). (12) LOBA Chemie, Mumbai, India: Papain Extra Pure (Cat. no. 05110). (13) Amresco LLC, Solon, OH, USA: Triton X-100 (Cat. no. Am-O694). (14) Synaptic Systems GmbH, Goettingen, Germany: Guinea pig polyclonal anti-VGLUT2 antibody (Cat. no. 135 404). (14) Affinity Biosciences, Cincinnati, OH, USA: Rabbit polyclonal anti-HEXB antibody (Cat. no. DF3074).

## 3. Results

### 3.1. Comparison of Population Composition of Cultures Prepared Using Trypsin and Papain

To determine the population composition of cell cultures obtained using trypsin (TRY culture) and papain (PAP culture), we initially used an immunocytochemical staining method using primary antibodies to neuron-specific enolase (NSE), glial fibrillar acid protein (GFAP) and Iba1, which are markers of neurons, astrocytes, and microglia, respectively. We evaluated the number of cells in 2, 7, and 14 DIV cultures. Because GFAP is a structural filament protein accumulating in the processes [25] and on the second day of cultivation, as our experiments have shown (see Appendix A), only a portion of astrocytes have processes (processes are removed during enzymatic treatment and subsequent mechanical dissociation during culture preparation), we hypothesized that counting cells based on immunocytochemical staining data, especially astrocytes, on the second day of cultivation might lead to incorrect results. In this regard, we present counting results only for 7 and 14 DIV cultures (Figure 1).

Immunostaining shows that the difference between the number of neurons and astrocytes in TRY and PAP cultures is negligible both in 7 and 14 DIV cultures (neurons—7 DIV *p* value: 0.06; 14 DIV *p* value: 0.076; astrocytes—7 DIV *p* value: 0.09767; 14 DIV *p* value: 0.07751). In turn, the amount of Iba1^+^ microglia (cells stained with anti-Iba1 antibodies) differed significantly (7 DIV ** *p* value: 0.0035; 14 DIV * *p* value: 0.0359).

To confirm the immunocytochemical staining data and indirectly assess physiological activity, we further characterized the number of cells of three populations by fluorescent imaging. As noted in Section 2, potassium chloride and ATP applications were used to identify neurons and glial cells (Figure 2A,B). Since two populations of glial cells in hippocampal cultures capable of responding to ATP are astrocytes and microglia [26] (oligodendrocytes do not proliferate under the culture conditions used [27]), we used vital dye CDr20 to identify microglia cells. Figure 2 shows an example of staining of 7 DIV PAP cultures. Experimentally, taking into account the recommendations of the developers and manufacturers, we have determined that the most optimal for staining in terms of signal-to-noise ratio in our conditions is the working dye concentration of 80 nM (microglial staining was noted even at lower concentrations, see Appendix A). According to CDr20 staining, ATP-responding CDr20^+^ cells were considered microglia, while CDr20^−^ cells responding to ATP were considered astrocytes. It should be noted that the patterns of responses to ATP in microglia and astrocytes differ (Figure 2C), which can be an additional criterion for the identification of these cell populations. Using calcium response patterns for KCl and ATP and the presence of CDr20 staining, we evaluated the number of neurons, astrocytes, and microglial cells in rat hippocampal PAP and TRY cultures. Figure 2D shows that, as with immunocytochemical staining, the number of microglia in PAP cultures is higher than in TRY cultures. A significant difference was noted in the case of 2, 7 DIV, and 14 DIV cultures (Interaction/Row Factor/Column Factor (DF, F (DFn, DFd), *p* value): 2, F(2, 38) = 0.1953, *p* = 0.8234/2, F(2, 38) = 11.58, *p* = 0.0001/1, F(1, 38) = 24.34, *p* < 0.0001).

It can be concluded that, according to two different methods of evaluating the population composition, the number of microglia cells in 7 DIV and 14 DIV PAP cultures significantly exceeds the number of microglia in TRY cultures, while differences in the number of neurons and astrocytes were not significant. Moreover, the number of microglia on the 7th day of cultivation increases compared to the second day, and this difference is most characteristic of cultures prepared using papain. Consequently, microglia in cultures are capable of proliferation even within the standard protocol of preparation and cultivation of neuron–glial cultures.

### 3.2. Papain Does Not Affect the Synaptogenesis and Maturation of Neuronal Networks

Synaptogenesis and maturation processes occur in neuronal networks in cultures, as in the intact brain. We did not note significant differences in the structures of emerging networks in PAP cultures compared to TRY cultures (Figure 1, Appendix A), and we found slightly higher neuronal content in PAP cultures on the second day of culture (Figure 2), although this difference was not significant in our experiments. However, the presence of neurons in cultures does not mean the presence of functional networks, so our further efforts were concentrated on testing functionality.

As is known, in neuronal networks between inhibition and excitation processes, there is a fine balance (E/I balance) where inhibition prevails over excitation. This balance is maintained through the regulation of the activity of excitatory glutamatergic and inhibitory GABAergic neurons. The presence of glutamatergic and GABAergic synapses, whose selective markers are vesicular transporters of glutamate and GABA (vGluT2 [28] and vGAT [29], respectively), can be considered an indirect confirmation of network formation by these neuronal subtypes. Figure 3A shows a representative image of a 14 DIV rat hippocampal PAP-neuron–glial culture stained with antibodies to vGAT and vGluT2. We used an antibody to the neuronal marker neuron-specific enolase (NSE) to visualize neurons. As shown in Figure 3A, PAP cultures demonstrate the formation of dense neuronal networks with developed GABAergic and glutamatergic innervation.

To confirm the presence of mature functional networks connecting hundreds of neurons in hippocampal PAP cultures, we used the calcium imaging method. Figure 3B shows the dynamics of [Ca^2+^]_i_ changes in representative neurons (red curves) of 14 DIV PAP culture. The presence of regular [Ca^2+^]_i_ oscillations occurring quasi-synchronously in multiple neurons in the field of view in the presence of GABA_A_R antagonist, bicuculline, may indicate the presence of a closed neuronal network with developed GABAergic innervation.

### 3.3. Microglia Morphology in PAP Cultures Differs from Microglia Morphology in TRY Cultures

Using dye CDr20, we found that microglia in the obtained neuron–glial cultures differed in shape and staining profile. Since significant differences in microglial quantity and minimal deviations in this parameter were noted above for 7 DIV cultures, further experiments were continued with cultures of this age. As for the morphology of microglia cells in cultures, like other researchers [30], we noted the presence of polarized microglia, which has mainly two opposite-sided processes (Figure 4A, upper row, purple arrow), as well as microglia with several processes diverging in different directions, which we defined as bushy (Figure 4A, upper row, yellow arrow). Similar morphological profiles were detected using anti-Iba1 antibodies (Figure 4A, lower row). Notably, the amount of polarized microglia (Figure 4B) was significantly higher in PAP cultures than in TRY cultures (Interaction/Row Factor/Column Factor (DF, F (DFn, DFd), *p* value): 1, F (1, 24) = 9.886, *p* = 0.0044/1, F (1, 24) = 0.4961, *p* = 0.4880/1, F (1, 24) = 23.47, *p* < 0.0001).

We also found interesting features of the loading/transformation kinetics of the fluorogenic UDP-glucoronosyl transferase substrate, CDr20, in the microglia cells of the two phenotypes. To evaluate loading kinetics, we performed staining with continuous fluorescence recording. The curves showing changes in CDr20 fluorescence in representative polarized (purple curves) and bushy microglial cells (red curves) are presented in Figure 4C. As a parameter most accurately highlighting the difference in cell loading kinetics patterns, we chose the time to reach 50% of maximum fluorescence intensity (half-rise time). As shown in the diagram in Figure 4D, the half-rise time is significantly higher in polarized microglia (F, DFn, Dfd, *p* value: 7.727, 39, 39, *p* < 0.0001). Consequently, dye transformation occurs significantly faster in bushy microglia. Such differences in transformation kinetics of the fluorogenic UDP-glucuronosyl transferase substrate may indicate, for example, a greater quantity or activity of this enzyme in bushy microglial cells.

Thus, the use of papain for hippocampal tissue processing promotes the preservation and proliferation of microglia in the neuron–glial culture with a predominantly polarized form.

### 3.4. Effect of the Combination of TGFβ+MCSF+Cholesterol Factors on the Population Composition of PAP and TRY Cultures Obtained Using Papain and Trypsin

The work published earlier demonstrated [10] that the amount of Iba1^+^ microglia in neuron–glial culture can be increased by adding the following combination of factors to the culture medium: TGFβ (2 ng/mL), IL-34 (100 ng/mL), and cholesterol (1.5 μg/mL). In our experiments, the IL-34 had no significant effect on the amount of Iba1^+^ microglia, which may be due to the use of hippocampal cells. Therefore, given the similarity of the action of IL-34 and macrophage-colony stimulating factor (MCSF) [31], we made an appropriate replacement. Thus, in our experiments, we introduced the following factors into the culture medium during the preparation of neuron–glial cultures: TGFβ (2 ng/mL), MCSF (5 ng/mL), and cholesterol (1.5 μg/mL). To compare the population composition of cultures grown with the addition of this combination of compounds, we used the above-mentioned calcium imaging approach with FluoriCa-8 staining, followed by the addition of KCl, ATP, and CDr20 loading.

TGFβ+MCSF+cholesterol results in an increase in the number of microglia in TRY cultures, so it was decided to study how this combination of compounds would affect the number of microglia in PAP cultures. The addition of factors was carried out during the preparation of hippocampal cell cultures, whereas the population composition was evaluated, as in the case of the previous series of experiments, in 2 DIV and 7 DIV cultures. Figure 5 shows representative images of 7 DIV cell cultures grown using this combination of factors.

As shown in Figure 5, the combination of factors contributes to an increase in the number of microglia in 2 DIV and 7 DIV cultures compared to cultures grown without the factors (Figure 2), both in the case of TRY cultures and in the case of PAP cultures. It should be noted that the morphology of microglia in the presence of TGFβ+ MCSF+cholesterol differs from that of microglial cells shown in Figure 2. In this case, both microphotographs with CDr20 distribution and transmitted light images show a large number of process-free round cells, which were not observed in the case of cultures grown without factors (Appendix A). These microglia cells have been designated by us as a separate group—ameboid microglia. In the case of PAP cultures, in addition to ameboid microglia, microglial cells with processes can be observed, indicating the heterogeneous population of this cell type. At the same time, as follows from the diagram in Figure 5, the number of neurons in TRY and PAP cultures varies insignificantly in 2 DIV and 7 DIV cultures (Interaction/Row Factor/Column Factor (DF, F (DFn, DFd), *p* value): 1, F (1, 20) = 0.2992, *p* = 0.5904/1, F (2, 38) = 0.4292, *p* = 0.5199/1, F (1, 20) = 0.01169, *p* = 0.9150). An interesting fact is that in the case of TRY cultures, phenotype change (prevalence of ameboid microglia) is already observed on the second day of culture (Figure 5C), as there is a significant decrease in bushy count and an increase in ameboid microglia count compared to PAP cultures (Interaction/Row Factor/Column Factor (DF, F (DFn, DFd), *p* value): 6, F (6, 60) = 114.2, *p* < 0.0001/3, F (3, 60) = 0.04526, *p* = 0.9871/2, F (2, 60) = 98.14, *p* < 0.0001). Ameboid microglia, as opposed to polarized and bushy, were also found to have a low-amplitude response to ATP application (Figure 5D, orange curve).

In the scratch assay test, we discovered (Figure 6) that amoeboid microglia have increased mobility, which can be judged by their migration to the damage site, and also, presumably, increased proliferative capacity, as their numbers significantly increase not only in the cut but also along its edges.

Thus, it can be concluded that the combination of TGFβ+MCSF+cholesterol factors results in both an increase in the number of microglia in cultures and their phenotypic changes, and in the case of TRY cultures, the phenotype change of a significant part of microglia occurs almost immediately after the culture preparation.

## 4. Discussion

In this work, we assessed the effects of trypsin and papain, used as enzymes for extracellular matrix dissociation, on the population composition of hippocampal cell cultures of newborn rats. According to the available conflicting literary data, the use of papain leads, on the one hand, to an increase in the number of neurons in cultures [14,32], whereas on the other hand, their morphology and probably network activity change [15]. In our experiments, we found that although the number of neurons in the first days of cultivation was slightly higher with papain, during further cultivation, their numbers did not significantly differ from cultures isolated using trypsin (Figure 1 and Figure 2). Our data do not contradict the literature in the context of population composition. We noted a slightly larger number of neurons in PAP cultures in early culture, whereas further offset of differences in neuronal count and the absence of pronounced changes in neuronal network morphology in our experiments may be due to differences in culture conditions and the subject of study. Using two different evaluation techniques, we have found that the number of microglia cells in PAP cultures is significantly different from the number of microglia in TRY cultures. It is known from literary data that papain is used as a proteolytic enzyme in the preparation of primary microglia cultures from the brain of both neonatal and adult animals [5,13], while its effect on microglial quantity in mixed neuron–glial cultures for developing a convenient model for neuroinflammation research has apparently not been assessed previously [4].

Although, depending on the microglia counting technique used, its number in our experiments varied almost twofold, the differences between PAP and TRY cultures were still reliable. Iba1 is a classic microglia marker used in a variety of neuroinflammatory studies [33]. However, according to a number of studies, its expression may vary depending on experimental conditions, and some researchers consider it a marker of active microglia [33,34]. Moreover, the existence of Iba1^−^ microglial populations in the brain [35,36,37], particularly in the hippocampus [35], has been previously reported. The selectivity of the UDP-glucuronosyl transferase fluorogenic substrate, CDr20, for microglial cells has been demonstrated both in vitro and in vivo [24]. It can be assumed that CDr20 can stain both Iba1^+^ and Iba1^−^ microglia. This assumption would explain the greater number of CDr20^+^ cells than Iba1^+^ microglia in cultures. In this regard, we hypothesized that in the case of mixed cultures, where microglia have a microenvironment consisting of neurons and astrocytes, Iba1 is an unreliable marker for assessing the total number of microglia. In future studies using more constitutive microglial markers, we plan to evaluate the ratio of CDr20^+^, Iba1^+^, and Iba1^−^ microglia and the possible overlap of these populations. Although the repertoire of microglial markers is quite extensive and includes many membrane-bound and cytosolic proteins, HEXb, TMEM119, and P_2_Y_12_ are considered constitutive, microglia-selective, and species-nonspecific [38,39]. However, for the latter two, expression changes depending on conditions have recently been demonstrated [40]. As our experiments showed, antibodies to HEXb stain the same cells as CDr20 (see Appendix A), confirming the greater reliability of this marker compared to Iba1 in mixed neuron–glial cultures.

The difference we found in CDr20 transformation kinetics between different microglia phenotypes can have two explanations: cells of different phenotypes differ in enzyme activity/expression or rate of fluorogenic substrate penetration into the cytosol. Regarding the level of expression, recently published work notes that UDP-glucoronosyl transferase, due to its stable expression under various conditions, can act as a universal marker of microglia, and the gene encoding it is a housekeeping gene [39]. Therefore, differences in the rate of CDr20 penetration can be considered the most likely explanation for the observed effect. Thus, dye transformation dynamics can be a criterion for phenotypic classification of microglia cells in vitro. In addition, the presence of ATP-induced [Ca^2+^]_i_ increase may be an additional criterion in this case. In the case of a combination of factors, the analog of which in earlier work was used to increase microglia proliferation in mixed neuron–glial cultures [10,11], we observed the appearance of a large amount of ameboid microglia characterized by a low-amplitude response to ATP application. Round amoeboid shape and changes in ATP response pattern can be considered manifestations of reactivity [41,42]. Changes in purinoreceptor expression profile during microglial activation, particularly decreased P_2_Y_12_ receptor expression and increased P_2_X_7_ receptor expression [43], which require more than 1 mM ATP for activation, were noted in other authors’ works, which may explain the decreased microglial sensitivity to ATP. Hence, microglia in cultures grown with TGFβ+MCSF+cholesterol can be considered predominantly activated. Indirect evidence of microglia reactivity is its mobility, as seen in our experiment with mechanical culture damage presented in Figure 6. Two days after scratching, in the case of cultures incubated with the combination of TGFb+MCSF+cholesterol factors, migration of amoeboid Iba1^+^ microglia to the damaged area was observed (Figure 6, bottom row). Such a phenomenon was not observed in cultures grown according to the standard protocol. Thus, a combination of features, including ameboid form, motility, significant proliferation, and low-amplitude ATP-induced Ca^2+^ response, suggests that TGFb+MCSF+cholesterol cause microglia activation in mixed neuron–glial hippocampal cultures. It is worth noting that the kinetics of CDr20 transformation in ameboid microglia cells had a similar pattern to the kinetics of bushy microglia transformation, so we do not show it separately in the figures.

An interesting feature of PAP cultures, in addition to increasing the number of microglia compared to TRY cultures, is the difference in its phenotype—namely, the greater quantity of polarized microglia, as well as the lesser quantity of amoeboid microglia in the case of culture incubation with TGFb+MCSF+cholesterol, at least on the second day of cultivation. If ameboid microglia are considered to be activated, it can be assumed that papain inhibits induced microglia activation. Similar effects of papain in suppressing proinflammatory signaling cascade activation have been previously demonstrated for other cell types [44,45,46,47]. In turn, if bushy microglia—for which more intense CDr20 transformation has been shown—is also considered active [34], but with an activation state that is probably not polar as in the case of amoeboid microglia, their reduced content in PAP cultures could also be evidence of papain’s inhibitory (anti-inflammatory) action regarding microglial activation. However, in this case, the activation is caused by manipulations during culture preparation. Similar astrocyte activation, caused by peculiarities of pure astrocytic culture preparation, has been demonstrated previously by other researchers [9].

## 5. Conclusions

Thus, it can be concluded that the use of papain as an enzyme for extracellular matrix degradation provides the formation of neuron–glial cultures with developed neuronal and astrocytic networks, as well as with more proliferative microglia compared to cultures prepared using trypsin. Such cultures can be fully considered mixed tricultures in which three major cell populations are present, with microglia in a more natural microenvironment. Unlike co-cultures of microglia and astrocytes or neurons, which involve isolating cultures of separate cell populations and their subsequent mixing [4], the approach used by other researchers and us [10,11] makes it possible to simplify in methodological terms the preparation of mixed cultures using biomaterial from one animal while avoiding substantial disproportion in population composition.

The observed differences in the quantity of Iba1^+^ and CDr20^+^ microglia in PAP and TRY cultures necessitate further evaluation of the reliability of using Iba1 as a microglial marker in mixed cultures. In turn, the demonstrated features of CDr20 transformation dynamics and patterns of calcium responses to ATP in microglial cells of different phenotypes can be used to classify this cell type. Although we showed that the quantity of microglia in PAP cultures exceeds that in TRY cultures, the question of microglial maturity in PAP cultures remains open and is the subject of further separate research. The existing method of assessing microglial maturity by its ability to induce caspase-3-mediated apoptosis when adding lipopolysaccharide (LPS) is rather indirect, considering that the TLR4 receptor, which is the target of LPS, can also be expressed by astrocytes [48,49], and as is known, astrocytes, like microglia, can release various pro- and anti-inflammatory cytokines involved in the implementation of the inflammatory response [50]. For comprehensive determination of microglial maturity, a combination of approaches will be required, including analysis of the expression of a specific set of genes, as well as immunocytochemical staining and morphometric analysis.

Also, the question of the reasons for the more intensive proliferation of microglial cells in PAP cultures remains open. It is known that for the survival of microglial cells both in vitro and in vivo, the activation of signaling cascades associated with TGFβ and CSF-1R [51] is necessary; however, as our experiments have shown, exogenous addition of modulators of these signaling pathways leads to excessive proliferation of microglial cells in vitro. One of the reasons for greater microglial proliferation in PAP cultures may be the peculiarity of the formed microenvironment, namely the architecture of neuron–glial networks. It is known that the microenvironment in the form of neurons and astrocytes positively affects microglial cells [52]. Furthermore, since MCSF does not cross the blood–brain barrier [53], it is secreted locally and can be synthesized by astrocytes [54]. Therefore, precisely the influence of the microenvironment through the secretion of necessary factors may determine the greater survival of microglia in PAP cultures. Thus, in the context of mixed cultures, the question of proliferation and properties of microglia may go beyond microglial biology and concern aspects of neuron–glial interaction as a whole. Additional comprehensive studies are required to answer this question.

## Figures and Tables

**Figure 1 brainsci-15-00442-f001:**
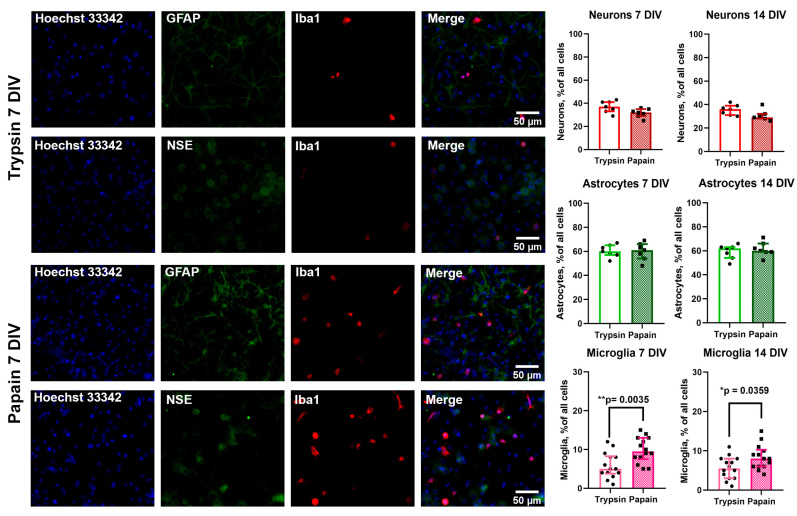
On the left: representative images of 7 DIV neuron–glial rat hippocampal cultures stained with antibodies to neuronal marker NSE (neuron-specific enolase, dilution 1:200), a marker of astrocytes GFAP (glial fibrillar acid protein, dilution 1:300), and calcium-binding protein Iba1 (ionized calcium-binding molecule 1, microglia marker; dilution 1:200). On the right: the diagrams showing the number of neurons, astrocytes, and microglia in 7 and 14 DIV TRY and PAP cultures. Mann–Whitney test. N = 7 for panels with astrocytes and neurons, N = 14 for panels with microglia. Each dot represents the mean number of cells in individual cell culture preparations, averaged by 4–5 viewfields. *p* < 0.05 (*), *p* < 0.01 (**).

**Figure 2 brainsci-15-00442-f002:**
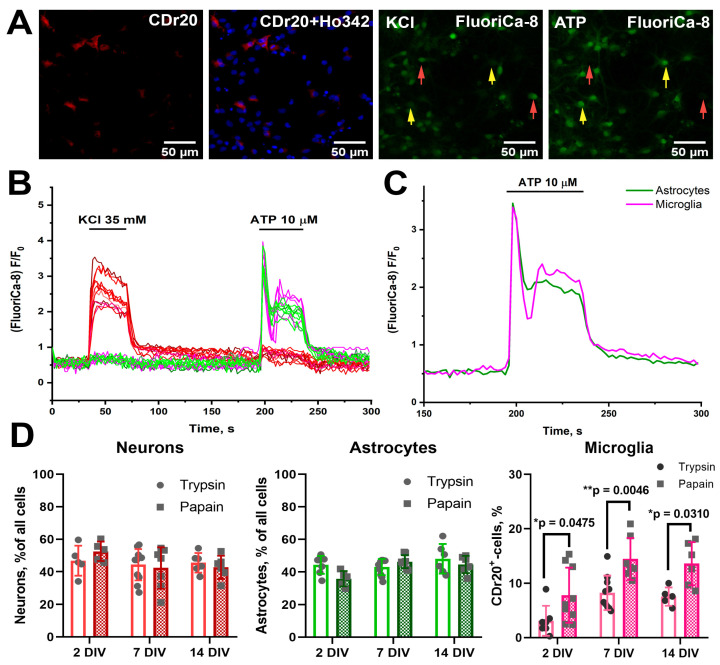
(**A**) Representative images of the rat hippocampal 7 DIV PAP culture region. Cell cultures were stained with a fluorescent Ca^2+^-sensitive FluoriCa-8 probe and vital microglia dye CDr20. Nuclei were stained with Hoechst 33342. In the case of FluoriCa-8, images corresponding to the time of KCl (35 mM) application with bright neurons and the time of ATP (10 μM) application with bright glial cells are shown. Representative neurons and glial cells are marked with red and yellow arrows, respectively. (**B**) Curves reflecting [Ca^2+^]_i_ changes in neurons (red curves), astrocytes (green curves), and microglia cells (purple curves) during KCl and ATP application. (**C**) Averaged astrocyte and microglia responses to ATP application. (**D**) Diagrams showing the number of neurons, astrocytes, and microglia in 2, 7, and 14 DIV cultures obtained using trypsin (columns without pouring) and papain (hatched columns). Two-way ANOVA followed by Sidak’s multiple comparisons test. N = 7. Each dot represents the mean number of cells in individual cell cultures (viewfields). *p* < 0.05 (*), *p* < 0.01 (**).

**Figure 3 brainsci-15-00442-f003:**
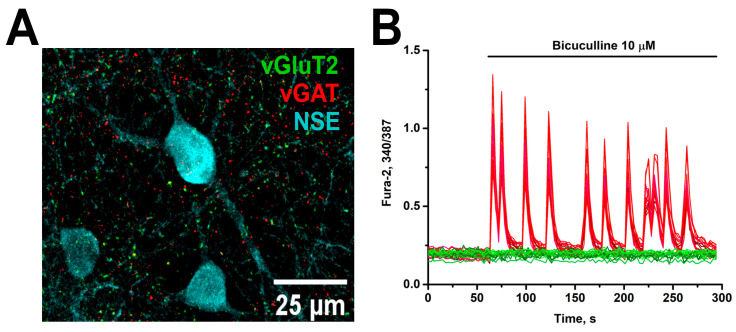
(**A**) Representative image of immunocytochemical staining of 14 DIV PAP culture with antibodies against glutamatergic synapse marker (vGluT2, dilution 1:500), GABAergic synapse marker (vGAT, dilution 1:500), and neuronal marker, neuron-specific enolase (NSE, dilution 1:200). N = 5; 4–5 individual viewfields were analyzed in each cell culture preparation. (**B**) Induction of regular high-amplitude [Ca^2+^]_i_ oscillations in neurons (red curves) of PAP cultures after application of GABAAR antagonist, bicuculline (10 μM). A ratiometric fluorescent Ca^2+^-sensitive Fura-2 probe was used to evaluate the [Ca^2+^]_i_ changes. Green curves correspond to glial cells. N = 5, n = 600 (100 neurons and 20 astrocytes in each experiment).

**Figure 4 brainsci-15-00442-f004:**
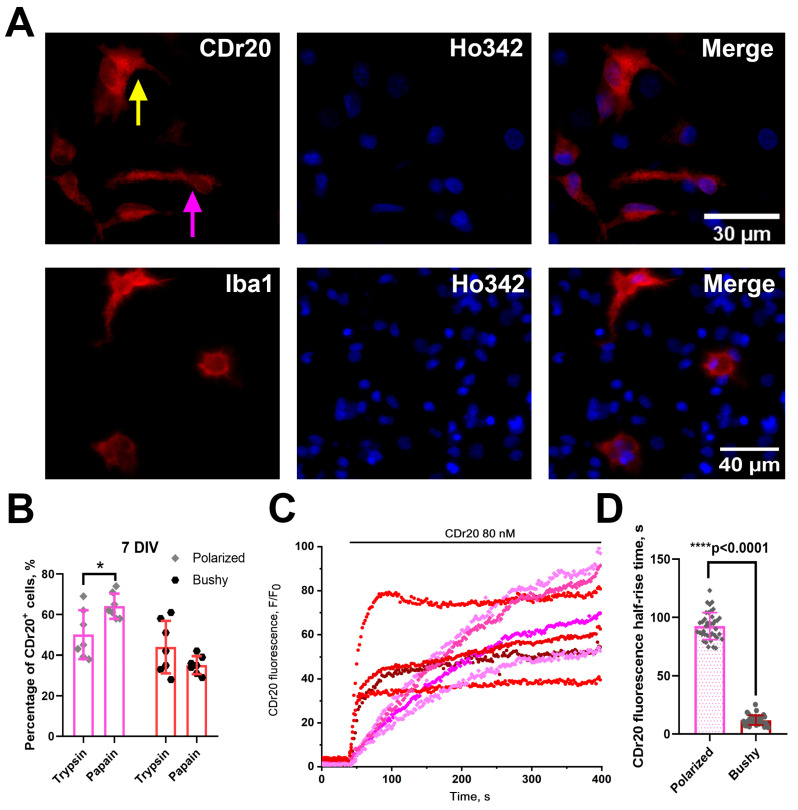
(**A**) Representative enlarged images of microglia in 7 DIV PAP cultures. Top row: CDr20^+^ microglia; bottom row: cultures stained with antibodies to Iba1. Yellow arrow shows a representative bushy microglial cell, while purple arrow shows a polarized microglia cell. (**B**) Diagram showing the amount of polarized (purple columns) and bushy microglia (red columns) in 7 DIV TRY and PAP cultures. Two-way ANOVA followed by Sidak’s multiple comparison test. N = 7; each dot corresponds to mean number of cells in individual cell culture, averaged by 4–5 viewfields. *p* < 0.05 (*). (**C**) Transformation kinetics of the fluorogenic UDP-glucoronosyl transferase substrate, CDr20 (microglia marker), in polarized (purple curves) and bushy (red curves) microglia. (**D**) A diagram reflecting differences in time to reach 50% of the maximum intensity of CDr20 fluorescence (half-rise time) in polarized (purple column) and bushy microglia (red column). Unpaired *t*-test. N = 4, n = 40. *p* < 0.0001 (****).

**Figure 5 brainsci-15-00442-f005:**
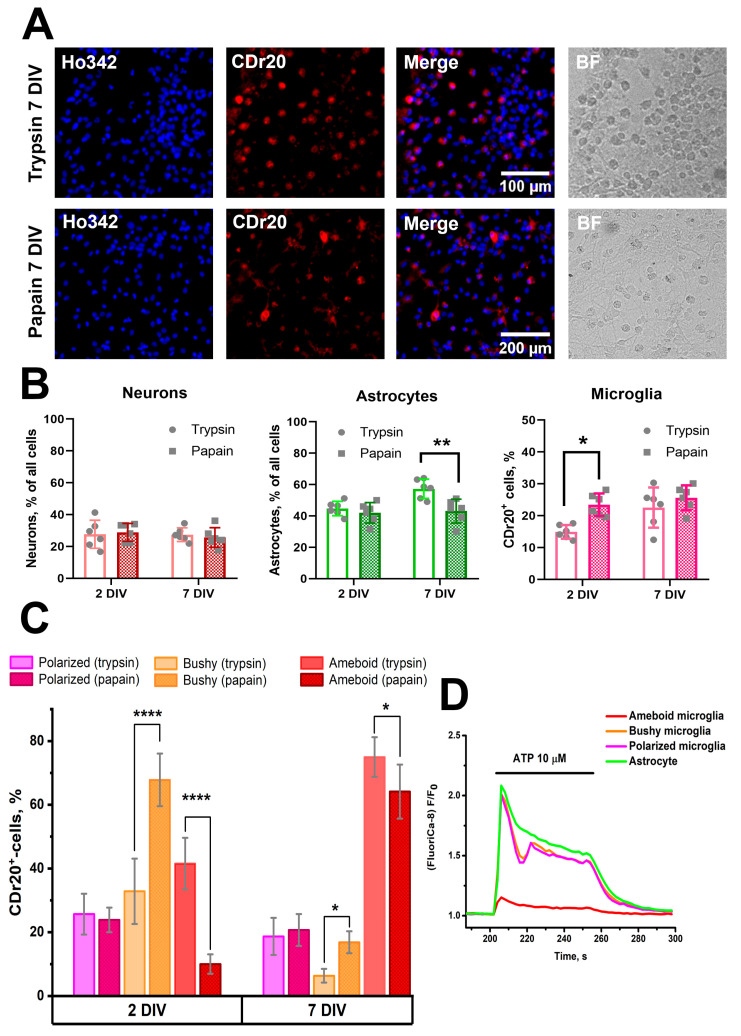
(**A**) Hippocampal TRY and PAP cultures grown in medium with addition of TGFβ+MCSF+cholesterol. Cultures were stained with a specific microglia marker, CDr20, as well as nuclei dye, Hoechst 33342. Additionally, images of the same area of culture in transmitted light (BF) are presented. (**B**,**C**) Diagrams showing the population composition of TRY and PAP cultures (**B**) and the microglia morphology (**C**) in 2 DIV and 7 DIV cultures grown in the presence of TGFβ+MCSF+cholesterol. In Diagram B, the PAP cultures correspond to hatched columns. N = 7; each dot corresponds to mean number of cells in individual cell culture, averaged by 4–5 viewfields. Two-way ANOVA followed by Sidak’s (**B**) or Tukey’s (**C**) multiple comparison test. *p* < 0.05 (*), *p* < 0.01 (**), *p* < 0.0001 (****). (**D**) Averaged curves showing [Ca^2+^]_i_ changes in astrocytes and microglia cells of different phenotypes during ATP application (10 μM).

**Figure 6 brainsci-15-00442-f006:**
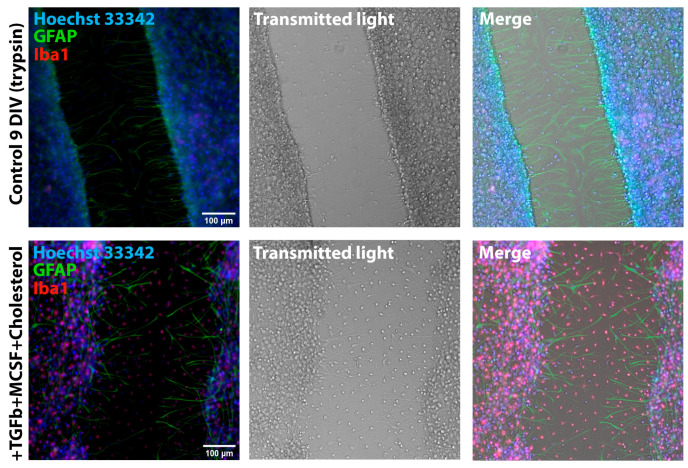
Images of 9 DIV TRY cultures grown without addition (upper row) and in the presence (lower row) of TGFβ+MCSF+cholesterol two days after scratching. Cultures were stained with antibodies against GFAP and Iba1. The nuclei were stained with Hoechst 33342.

## Data Availability

The original contributions presented in the study are included in the article’s Appendix A. Further inquiries can be directed to the corresponding authors.

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
