# Peer review of "Papain Affects the Percentage and Morphology of Microglia in Hippocampal Neuron–Glial Cultures"

_brainsci, 2025, doi:10.3390/brainsci15050442_

Round 1

Reviewer 1 Report

Comments and Suggestions for Authors

Among the studies, the used trypsin concentrations can vary. Did the authors also test other concentrations?

Based on other studies (e.g., Kim B, Fukuda M, Lee JY, et al. Visualizing Microglia with a Fluorescence Turn-On Ugt1a7c Substrate. Angew Chem Int Ed Engl. 2019;58(24):7972-7976.), Glucuronidation of CDr20 by Ugt1a7c in microglia produces a bright red fluorescence consistent with the expression of the markers P2ry12, Csf1r, Cx3cr1 and Iba-1. But the authors found inconsistent results between CDr20 and Iba1. Additional experiments with other microglial markers are necessary.

Please follow the recommendations on the use of microglial nomenclature: Paolicelli RC, Sierra A, Stevens B, et al. Microglia states and nomenclature: A field at its crossroads. Neuron. 2022;110(21):3458-3483. doi:10.1016/j.neuron.2022.10.020

The Results Section needs to be edited, because there are several extern references. Some information can be moved to introduction or discussion.

Author Response

Reviewer 1

The authors sincerely thank the Reviewer for the comments provided and for the very useful reference to work with a new approach to describing microglial nomenclature.

Comments 1: Among the studies, the used trypsin concentrations can vary. Did the authors also test other concentrations?

Response 1: Yes, in earlier studies we tested various concentrations of trypsin, including a solution with a concentration of 0.25%, which is most commonly used in research. However, in this case, the number of isolated cells, particularly neurons, is significantly lower (for example 10.4103/1673-5374.244794; 10.1016/j.ebiom.2019.08.050), and a large amount of debris is observed, which reduces the overall survival of cultures. The smaller number of neurons does not allow the formation of mature networks in which spontaneous or inducible synchronous activity is possible, which is an important characteristic of cultures when modeling and studying the mechanisms of neuronal activity in brain networks. In studies that used low concentrations of trypsin, typically very sparse cultures are used. In turn, when using a concentration of 1%, as in all our previous works (10.21769/BioProtoc.5199; 10.1016/j.abb.2024.109951; 10.1111/jnc.15729;  10.1002/glia.23763), we have the ability to obtain 12-15 dishes with cell cultures from one animal, which allows for the most rational use of biomaterial.

Comments 2: Based on other studies (e.g., Kim B, Fukuda M, Lee JY, et al. Visualizing Microglia with a Fluorescence Turn-On Ugt1a7c Substrate. Angew Chem Int Ed Engl. 2019;58(24):7972-7976.), Glucuronidation of CDr20 by Ugt1a7c in microglia produces a bright red fluorescence consistent with the expression of the markers P2ry12, Csf1r, Cx3cr1 and Iba-1. But the authors found inconsistent results between CDr20 and Iba1. Additional experiments with other microglial markers are necessary.

Response 2: For the listed markers, as noted by other researchers and as we note in the discussion, changes in expression have been shown depending on conditions, including the P2Y12 receptor, considered a constitutive marker of microglia. In this regard, we used the HEXB marker (10.1038/s41590-020-0707-4), for which no changes in expression depending on conditions have been demonstrated so far. The figure below (please, see the attached file) shows an image demonstrating the staining of 7DIV PAP-culture with CDr20 and staining of the same culture area with antibodies to HEXB. The alignment of images before and after fixation was performed using a marker grid applied to the bottom of the coverglass with cultures, as described previously (10.1002/glia.23763). As the Figure shows, CDr20 and HEXB stain the same cells in the view field. In negative control experiments, we have found that CDr20 completely leaks from the cells during fixation and permeabilization of cells. This figure will be added to the Supplementary and these experiments will form the basis for our further work.

Comments 3: Please follow the recommendations on the use of microglial nomenclature: Paolicelli RC, Sierra A, Stevens B, et al. Microglia states and nomenclature: A field at its crossroads. Neuron. 2022;110(21):3458-3483. doi:10.1016/j.neuron.2022.10.020

Response 3: The authors thank the reviewer for the provided reference. We will certainly use the specified classification approaches in further work. We deliberately avoided the binary M1/M2 classification, understanding its limitations. As for "reactivity," in this case we also limited the use of this term, since it is currently unclear what microglia in cultures grown in the presence of TGF+MCSG+cholesterol could be reactive to, if cells exhibit properties of traditionally reactive microglia without any inducers, such as LPS. Therefore, in the context of this work, we chose a descriptive approach in terms of morphology and properties, as the work is a starting point for our further research, including the use of CDr20 as a marker. Also, like some other researchers, we avoided the term "subpopulation" due to the vagueness of criteria for division specifically into populations.

Comments 4: The Results Section needs to be edited, because there are several extern references. Some information can be moved to introduction or discussion.

Response 4: We have corrected the description of the results. Some paragraphs, particularly regarding the description of calcium dynamics, have been removed. The remaining references and explanations are necessary to explain the need for certain experiments.

Reviewer 2 Report

Comments and Suggestions for Authors

The manuscript titled "Papain affects the percentage and morphology of microglia in hippocampal neuron-glial cultures" investigates a critical methodological aspect of neuroscience research—how different proteolytic enzymes (papain vs. trypsin) influence microglial populations in hippocampal neuron-glial cultures. The authors provide compelling evidence that papain-prepared cultures (PAP-cultures) contain a significantly higher proportion of microglia compared to trypsin-prepared cultures (TRY-cultures) while maintaining a stable number of neurons and astrocytes. The study also reports distinct morphological differences, with PAP-cultures exhibiting more polarized microglia and TRY-cultures showing a higher prevalence of bushy microglia. Furthermore, the authors demonstrate that the addition of TGFβ, MCSF, and cholesterol enhances microglial proliferation and induces the appearance of amoeboid, reactive microglia. The use of CDr20, a microglia-specific fluorescent probe, in combination with calcium imaging and immunostaining, strengthens the study’s findings. However, several methodological and conceptual concerns need to be addressed to enhance the robustness and applicability of the results.

Comments:

1. The study claims that papain increases microglial populations without affecting astrocytes or neurons. However, quantification of additional neuronal and astrocytic markers beyond NSE and GFAP (e.g., NeuN for neurons, S100β for astrocytes) would provide stronger validation.

2. While morphological analysis suggests that microglia in TRY-cultures are more reactive, functional assays such as cytokine profiling (IL-1β, IL-6, TNF-α) or phagocytosis assays should be included to confirm their activation state.

3. The reliance on morphology alone to infer microglial activation status is a limitation. Markers such as CD68 or Ki67 should be assessed to confirm microglial activation and proliferation.

4. The rationale for selecting hippocampal cultures is not fully justified, given that microglia are enriched in cortical regions. Were other brain regions, such as the cortex or cerebellum, considered for comparative analysis?

5. Since the authors suggest that PAP-cultures better mimic in vivo conditions, it is recommended to validate these findings in an in vivo setting, such as murine models, to rule out species-specific effects.

6. The impact of enzymatic digestion duration is not addressed. Since papain requires a longer incubation time (25 min vs. 10 min for trypsin), has the total cellular yield been compared between the two conditions?

7. The study focuses on DIV 2, 7, and 14. However, most mechanistic studies use neurons at DIV 21. Have the authors analyzed neuron-glial interactions at DIV 21 to assess long-term functional and morphological changes?

8. As microglia play a key role in neuroinflammation, it would be relevant to expose cultures to an inflammatory stimulus (e.g., LPS treatment) to compare the inflammatory response in TRY- and PAP-cultures.

9. The authors suggest that PAP-cultures lead to more physiologically relevant neuron-glial interactions. However, astrocytes are the most abundant CNS cell type. Can the authors clarify how astrocytic populations remain balanced in these cultures?

10. While the study shows that papain does not alter neuronal response to KCl, confirming neuronal excitability through electrophysiological recordings (patch-clamp) would strengthen the claim.

11. Since trypsin-prepared cultures exhibit fewer microglia, apoptosis and necrosis markers such as Caspase-3, Bax, Bcl-2, and Annexin V should be analyzed to determine whether the decrease is due to increased cell death.

12. Despite an increase in microglial cells following TGFβ + MCSF + cholesterol treatment, neuronal numbers appear to decline. What could be the potential mechanism underlying this loss?

13. To ensure research transparency, details on the number of cultures prepared, the number of cells analyzed, and the number of images used for quantification should be included in figure legends.

14. The authors should comment on whether this papain-based neuron-glia culture method is applicable to mice, as only rat-derived cultures were used in this study.

15. Batch effects or variability between different culture preparations may have influenced the results. Were normalization methods applied to account for differences in total cell numbers between TRY- and PAP-cultures?

16. The study proposes that TGFβ, MCSF, and cholesterol drive microglial proliferation and activation. However, how do the authors rule out individual contributions of each factor? Additional controls for each component are necessary.

17. RNA sequencing should be performed to provide a broader perspective on gene expression differences between PAP- and TRY-cultures and to identify pathways underlying observed phenotypic changes.

18. The authors do not discuss potential limitations of their study. It would be beneficial to outline any methodological constraints and future directions.

19. A detailed flowchart of the experimental protocol should be included to facilitate replication by other researchers.

20. The type of culture dishes or plates used for triculture preparation should be specified, as different surfaces may influence cell adhesion and proliferation.

21. How was cell viability assessed following dissociation with trypsin and papain? Were differences in microglial numbers due to differential cell survival rather than actual proliferation?

22. Given that single-cell sequencing technologies are becoming more accessible, how do the authors anticipate their findings will contribute to future transcriptomic analyses of neuron-glia interactions?

23. The study does not provide mechanistic insights into how papain might exert anti-inflammatory effects on microglia. Further investigation is needed to support this claim.

24. The authors conclude that PAP-cultures better mimic the natural microglial environment. However, without direct comparisons to in vivo microglial transcriptomic profiles, this assertion remains speculative.

Comments on the Quality of English Language

The manuscript is generally well-written; however, there are areas where clarity, conciseness, and grammatical structure could be improved. Some sentences are overly complex and could benefit from simplification for better readability. Additionally, certain technical terms and methodological descriptions could be expressed more precisely to ensure clear communication. The manuscript would benefit from professional language editing to refine sentence structure, improve flow, and eliminate minor typographical errors. Careful proofreading is recommended to enhance clarity and coherence, particularly in the methods and discussion sections.

Reviewer 3 Report

Comments and Suggestions for Authors

Introduction: The authors should see "only in several studies (line 66)".

Results: The "Results" section should be more direct in describing the findings. Extensive discussions should be moved to the "Discussion" section. The authors should describe the F-statistic for the comparisons in the results.

It is necessary reference to this sentence: "We used a concentration of 10 μM, which is sufficient to activate various subtypes of P2Y and P2X receptors." lines 269-270.

Figure 6 should be in the "Results" section, with the appropriate discussion in the "Discussion" section.

Author Response

Reviewer 3

Comments 1: Introduction: The authors should see "only in several studies (line 66)".

Response 1: We thank the Reviewer for this remark. The sentence has been corrected.

Comments 2: Results: The "Results" section should be more direct in describing the findings. Extensive discussions should be moved to the "Discussion" section. The authors should describe the F-statistic for the comparisons in the results.

Response 2: According to the comments of Reviewer 1, we have removed from the results section the paragraphs describing the methods and mechanisms of the observed effects. The remained external references and sentences are necessary for explanation of the performed experiments. We have corrected Figure 1, since the results were analyzed with Mann-Whitney test in this case. For the results analyzed with parametric tests and discussed in the text of the manuscript, we have provided F-statistic parameters.

Comments 3: It is necessary reference to this sentence: "We used a concentration of 10 μM, which is sufficient to activate various subtypes of P2Y and P2X receptors." lines 269-270.

Response 3: The authors thank the Reviewer for this remark. Since this paragraph has been removed, we have included the necessary reference to Materials and methods section.

Round 2

Reviewer 2 Report

Comments and Suggestions for Authors

In this revised manuscript, instead of performing new experiments, the authors addressed most of the comments by providing literature references. The manuscript would have been more compelling if the authors had performed at least 50% of the suggested experiments.

Author Response

In response to the reviewers' comments, we have conducted several series of additional experiments. The comments provided by Reviewers 1 and 3 were specific and addressed the core aspects of our research. Based on the SuSy system, our revisions appear to have adequately addressed their concerns.

Regarding Reviewer 2, we have provided comprehensive responses to all comments, including additional experimental data, references to the text of the submitted manuscript, citations from our previously published works in peer-reviewed journals, and methodological articles. We note that several of Reviewer 2's comments did not request specific experimental work, but rather raised theoretical points for discussion. Furthermore, some suggested experiments involving RNA sequencing and in vivo models extend significantly beyond the scope of the present study. We find it particularly concerning that the reviewer requested in vivo experiments for a study explicitly focused on in vitro culture protocols, which suggests a potential misunderstanding of our research context and raises questions about the assessment's objectivity.

We respectfully disagree with the reviewer's apparent expectation that a single paper partially regarding microglia should encompass all possible experimental techniques. As clearly stated in our response, this work represents an initial step toward characterizing microglia in neuron-glial cultures in vitro. The comprehensive characterization recommended by Reviewer 2 would necessitate collaboration with multiple specialized research groups and would require substantially more time than is reasonable for addressing revision requests.

We appreciate your attention to this matter and look forward to your response.

Sincerely,
Sergei Gaidin